# Variations in Rates of Discharges to Nursing Homes after Acute Hospitalization for Stroke and the Influence of Service Heterogeneity: An Anglia Stroke Clinical Network Evaluation Study

**DOI:** 10.3390/healthcare8040390

**Published:** 2020-10-09

**Authors:** Michelle Tørnes, David McLernon, Max O Bachmann, Stanley D Musgrave, Diana J Day, Elizabeth A Warburton, John F Potter, Phyo Kyaw Myint

**Affiliations:** 1Ageing Clinical and Experimental Research (ACER) Group, Institute of Applied Health Sciences, School of Medicine, Medical Sciences and Nutrition, University of Aberdeen, Scotland AB24 3FX, UK; 2Medical Statistics Team, Institute of Applied Health Sciences, School of Medicine, Medical Sciences and Nutrition, University of Aberdeen, Scotland AB24 3FX, UK; d.mclernon@abdn.ac.uk; 3Norwich Medical School, University of East Anglia, Norwich NR4 7TJ, UK; M.Bachmann@uea.ac.uk (M.O.B.); S.Musgrave@uea.ac.uk (S.D.M.); John.Potter@uea.ac.uk (J.F.P.); 4Addenbrooke’s Hospital, Cambridge CB2 0QQ, UK; diana.day@addenbrookes.nhs.uk (D.J.D.); eaw23@medschl.cam.ac.uk (E.A.W.); 5Stroke Research Group, Norfolk and Norwich University Hospital, Norwich NR4 7UY, UK

**Keywords:** nursing home, institutionalization, acute hospitals, health services research, stroke, UK, NHS

## Abstract

Nursing home placement after stroke indicates a poor outcome but numbers placed vary between hospitals. The aim of this study is to determine whether between-hospital variations in new nursing home placements post-stroke are reliant solely on case-mix differences or whether service heterogeneity plays a role. A prospective, multi-center cohort study of acute stroke patients admitted to eight National Health Service acute hospitals within the Anglia Stroke and Heart Clinical Network between 2009 and 2011 was conducted. We modeled the association between hospitals (as a fixed-effect) and rates of new discharges to nursing homes using multiple logistic regression, adjusting for important patient risk factors. Descriptive and graphical data analyses were undertaken to explore the role of hospital characteristics. Of 1335 stroke admissions, 135 (10%) were discharged to a nursing home but rates varied considerably from 6% to 19% between hospitals. The hospital with the highest adjusted odds ratio of nursing home discharges (OR 4.26; 95% CI 1.69 to 10.73), was the only hospital that did not provide rehabilitation beds in the stroke unit. Increasing hospital size appeared to be related to an increased odds of nursing home placement, although attenuated by the number of hospital stroke admissions. Our results highlight the potential influence of hospital characteristics on this important outcome, independently of patient-level factors.

## 1. Introduction

Stroke is the second leading cause of mortality and the third leading cause of disability worldwide [1]. Given the significant impact that a stroke can have on the ability to perform daily tasks, many patients require institutionalization. In the UK, for example, 8.2% of patients surviving stroke are discharged to a care home [2]. However, changing demographics, improvements in stroke care and post-acute care provision (i.e., widespread incorporation of early supported discharge policies) have led to a significant reduction in institutionalization rates since 1995 [3].

Home discharge is the most desirable outcome following stroke; 85% of patients in a previous study favored returning home for reasons such as increased autonomy and privacy [4]. Furthermore, greater long-term improvements have been observed in patients that return home [5].

Understanding what determines a patient’s discharge destination after stroke can provide invaluable information to identify factors that can be modified to prevent the need for nursing home placement. This information is also important in ensuring efficient discharge planning. If discharge to a nursing home can be anticipated then communication between different health care providers can be initiated early, as well as discussions with the patient and their family. In UK hospitals, the decision as to the patient’s discharge destination following stroke is arrived at collaboratively following input from the patient, the patient’s family, the leading physician, social work, and other allied health professionals.

To date, most studies have focused on what patient characteristics drive the need for nursing home placement after stroke. Other important factors, such as differences in service provision and resources, may impact the need for institutionalization. However, few have researched this specifically, despite a wealth of research showing the importance of hospital factors such as staffing levels and hospital stroke volume on other outcomes [6,7,8,9,10].

The aim of this study, therefore, was to examine whether variations in discharge to nursing homes after stroke could, in part, be explained by heterogeneities in hospital-related factors.

## 2. Materials and Methods

### 2.1. Study Design

A multi-centre, prospective cohort study conducted at eight acute NHS hospitals within the Anglia Stroke and Heart Clinical Network (AS&HCN) which participated in the Anglia Stroke Clinical Network Evaluation Study (ASCNES). The network covers three counties in the East of England, UK, with a catchment population of approximately 2.5 million. The ASCNES primary outcome of interest was one-year mortality, whereas discharge destination, the focus of this paper, was one of the secondary outcomes of interest. The detailed study protocol has been previously published [11].

### 2.2. Participants

The study population included all patients, aged 18 years or older, admitted to any of the eight hospitals (which spanned both urban and rural regions) within the ASCNES group diagnosed with stroke by an accredited stroke physician. Diagnoses by the stroke physician were coded using International Classification of Diseases-10 (ICD-10) and were confirmed through cerebral imaging. Stroke was defined as a focal neurological impairment of sudden onset and lasting more than 24 h (or leading to death) because of an intracerebral ischemic or hemorrhagic event. This definition was used in order to exclude diagnoses of transient ischaemic stroke, subdural hematomas, and subarachnoid hemorrhages. The study sample was systematically selected to include all patients with stroke admitted every third month of a two-year period (October 2009 to September 2011), resulting in a total of eight study months; the robustness of this sampling technique has been verified [12]. Patients who were previously resident in a nursing home prior to their admission for stroke, patients who died as an inpatient or those who were discharged to an interim or rehabilitation setting were all excluded from the final study population.

### 2.3. Data Collection

This study made use of prospectively collected anonymized data that were routinely sent to the AS&HCN for performance assessment, in addition to baseline patient and outcome data retrieved from case records and discharge summaries by clinical teams. Any identifiable patient information was held only at the individual hospitals—the network and investigators did not have access to these details.

Data on hospital-level characteristics were collected from clinical leads or service managers at each hospital and updated every six months over the two-year study period by research staff [11].

### 2.4. Statistical Analyses

We performed a single-level multiple logistic regression model using maximum likelihood with the hospital as a fixed-effect. Our outcome measure was new nursing home placement at discharge from acute hospitalization for stroke amongst those who were not previously institutionalized. Cases discharged to a nursing home were treated as the event group, whilst those who were discharged back home (including those living in sheltered accommodation or retirement home) constituted the non-event group.

Patient risk factors of institutionalization following stroke were identified a priori from a literature review (see Appendix A). The distribution of the patient risk factors available in the dataset was examined between hospitals using hypothesis testing. Factors that were unevenly distributed amongst the hospitals (*p*-value set at 0.05 significance level) were included in the main model as they were considered potential confounders of the relationship between our independent variable (hospital) and outcome of interest (institutionalization). These included a patient’s pre-stroke modified Rankin Scale (mRS) score, discharge mRS score, acute hospital length of stay, and whether the patient had a total anterior circulation stroke (TACS), history of myocardial infarction or ischemic heart disease, or developed an inpatient complication (see Appendix A). For the interest of the reader, we provide Appendix A with a breakdown of the different types of complications recorded and the proportion of patients that were reported to have developed them. Furthermore, although we identified stroke severity as an important risk factor, 73% of cases had missing data on The National Institutes of Health Stroke Scale (NIHSS) score, and we believe that this variable was only collected in patients that were eligible for thrombolysis. Therefore, we did not multiply impute this variable due to the potential for introducing information bias. Instead, we adjusted for proxy variables of stroke severity, such as discharge mRS score [13], whether the patient had a TACS [14,15], or showed symptoms of brain lateralization. We hypothesized that patients with no symptoms of brain lateralization are likely to be those with more severe stroke where the whole brain is affected.

Processes of care measures such as the delivery of acute treatment therapies were not accounted for in our study as we believe they are mediator variables that lie on the causal pathway between hospital-level factors and stroke patient outcomes [16]. Their inclusion in our regression model could otherwise lead to over-adjustment bias by obscuring the overall effect of the hospital [17,18].

To explore hospital-level factors we first compared, descriptively, service variations between hospitals. We also plotted the adjusted odds ratio of new nursing home placement (estimated from the regression model) against the hospital characteristics of interest for each hospital. This is the favored approach when the sample size of hospitals is below the suggested critical number required to reliably estimate hospital effects through multi-level modelling [19,20]. Hospital characteristics included hospital size (number of hospital beds), hospital type (secondary or tertiary), hospital stroke volume (number of stroke admissions per month), presence of vascular surgery onsite, distance to a neurosurgical facility, number of full-time-equivalent (fte) staff per five beds (including doctors, nurses, occupational therapists, speech and language therapists, physiotherapists and dieticians), number of stroke unit beds per 100 admissions, number of hospital beds per CT scanner, number of non-stroke patients treated on the stroke unit per five beds, and number of patients with stroke treated outside the stroke unit per five beds.

The index of multiple deprivation (IMD) score was also explored graphically to assess whether the socio-economic status of the hospital catchment area may explain any differences in outcome observed between hospitals. This is an aggregate measure and is based on several domains, including income, employment, education, health, crime, barriers to housing and services and the living environment, that are believed to provide an indication of deprivation. England is subdivided into 32, 844 smaller areas, with an IMD score of 1 representing the area in England that is considered to be the most deprived and a score of 32, 844 the least deprived [21]. Here we assigned the mean 2010 IMD scores of the areas that make up the counties of Suffolk, Norfolk and Cambridgeshire and assigned these to each of the hospitals to which they are located [22].

We carried out multiple imputations by chained equations using the Multivariate Imputation by Chained Equations (MICE) package in R and pooled the results using Rubin’s rules [23] (see Appendix A for details on variables used to inform multiple imputations). The inclusion of auxiliary variables for the imputation increased the casewise missingness to 43%, thus 43 datasets were imputed. The distribution of sample characteristics between individuals with complete and incomplete data was compared using the appropriate statistical tests (Appendix A). Findings from the complete case analysis are presented in Appendix A.

### 2.5. Sensitivity Analyses

We performed two sensitivity analyses to check the robustness of our findings. The first of these included all patient risk factors identified a priori in a literature review that were available in our dataset irrespective of whether they were evenly distributed between hospitals or not, and the second excluded hospital 2 which did not contribute data after January 2010.

All analyses were performed using R version 3.3.1 (R, Vienna, Austria) for Windows [24].

### 2.6. Ethics Approval

Ethical approval was obtained from the NRES Committee East of England—Norfolk (REC Reference number 10/H0310/44).

## 3. Results

Of the 2656 patients admitted during the inclusion period, 179 were eventually diagnosed with a condition other than stroke. After exclusions (see Appendix A), a total of 1335 cases were available for analyses (median age (IQR) 79 (70 to 86 years), 52% males, 83% ischemic stroke).

A total of 135 (10%) patients were placed in a nursing home at discharge, with the remaining 90% discharged home. The proportion of patients discharged to a nursing home varied between hospitals (6% in hospital 1 to 19% in hospital 3). The unadjusted odds of being discharged to a nursing home in hospital 3 were 3.84 times that of hospital 1; the highest odds of all hospitals in the study (*p* < 0.001). All other hospitals were associated with greater odds of nursing home placement than hospital 1, although none reached statistical significance.

To note, when examining the original study population (including those who were discharged to settings other than home or nursing homes) the hospitals with the lowest rates of nursing home placements were not the hospitals with the highest proportion of patients discharged to an interim or rehabilitation setting, or who died as an inpatient (Appendix A).

Between-hospital variations in rates of nursing home placement remained after the adjustment of patient confounders (Table 1). Hospital 3 still had the highest odds of discharges to nursing homes (Odds Ratio (OR) 4.26; 95% confidence interval (CI) 1.69 to 10.73). Due to low event numbers in the other hospitals, none of these were statistically significantly different from hospital 1, and all had wide confidence intervals. Findings were similar for the complete case analysis (Appendix A) and sensitivity analyses (Table 2).

Table 3 outlines the service characteristics of each hospital, including standardized staffing levels, bed capacity, and the availability of onsite facilities. There is notable heterogeneity in service provision and resource levels. For example, only hospital 1 had a neurosurgical facility onsite and only hospitals 1, 3, 5 and 6 had onsite vascular surgery.

Hospital 3, with the highest odds ratio, and the only hospital that showed a statistically significant difference compared to hospital 1, was the study’s largest secondary hospital and was the only hospital that did not have any rehabilitation beds within the stroke unit. It also had the highest number of junior doctors and occupational therapists per five beds but one of the lowest number of nurses per five beds. Compared to hospital 1, hospital 3 was smaller, had a lower volume, had less stroke unit beds per 100 admissions, fewer hospital beds per CT scanner, had more senior doctors, junior doctors, physiotherapists and occupational therapists per fives beds, had fewer nurses, speech and language therapists and dieticians per five beds, less non-stroke patients treated on the stroke unit and fewer patients with stroke treated outside the stroke unit.

No obvious patterns emerged for any of the hospital characteristics in our graphical analysis (Appendix A). However, when we plotted hospital size against the adjusted odds ratio of nursing home placement, and grouped by hospital stroke volume, there was a clear positive linear trend in point estimates, although confidence intervals were wide. The increased odds ratio with hospital size was much more attenuated in the group of hospitals that admitted more than 50 patients with stroke a month (see Figure 1).

## 4. Discussion

Considerable variations in rates of new nursing home placements after stroke exist between hospitals in the UK NHS setting even after the adjustment of case-mix differences. These variations cannot be explained in terms of differences in rates of discharges to interim or rehabilitation settings, or inpatient deaths. Instead, they appear to reflect heterogeneities in service provision and resource levels between hospitals. Our exploratory analyses indicated the potential importance of the provision of rehabilitation beds within the stroke unit, hospital size and volume.

Our main finding agrees with a previous study conducted across health regions in the UK [25]. Here, we suggest that differences in hospital characteristics may partly explain these between-hospital variations. The superiority of stroke rehabilitation that is provided within a comprehensive stroke unit (i.e., where patients receive acute treatment and rehabilitation in combination) compared to rehabilitation provided outside the unit has been previously demonstrated [26,27,28]. Therefore, we surmise that this may be one driving force behind the high rate of nursing home placements in hospital 3 as this was the only hospital that did not provide rehabilitation beds in the stroke unit. It is unclear why this would have an effect on discharge destination; although it has been suggested that early mobilization and rehabilitation on these wards may lead to better functional gains in patients, allowing them to safely return home [28], a recent systematic review was unable to find conclusive evidence that early mobilization leads to improved outcomes after stroke [29].

This study also highlights the potential importance of hospital size and the volume of stroke admissions. In addition to hospital 3 being the study’s largest, secondary hospital, our graphical exploration demonstrated a strong positive trend between hospital size and our outcome, for low volume hospitals. This finding may reflect heightened resource pressure in larger hospitals. Indeed, out of all the other secondary hospitals, hospital 3 had the highest ratio of hospital beds to CT scanners. Pressure on CT scanners may lead to a delay in a confirmatory diagnosis and treatment, subsequently hindering recovery and making a safe discharge home less probable [30].

Larger patient volumes have been linked to improved outcomes in stroke [6,10]. This may be because physicians in high volume hospitals gain more experience in dealing with certain conditions and are therefore better placed at providing effective care [31,32]. Our finding that the effect of hospital size appears to be attenuated by hospital stroke volume appears to align with this and studies that have shown that whilst larger hospitals are associated with poorer outcomes, higher volume hospitals produce better results [33,34].

The main strength of our study is its prospective design which facilitated the collection of detailed and accurate patient and hospital-level data. This minimized residual confounding by case-mix differences and allowed us to explore several hospital characteristics. We did not restrict our analysis to certain sub-populations, meaning our results are generalizable to the wider stroke population. Multiple imputations also minimized any biases that would have arisen from the complete case analysis. Another strength was that our statistical approach was the most robust and economical way to use the data available. The sensitivity analyses conducted further confirmed the vigor of our findings. Very few other studies have looked at the influence of hospital characteristics on this outcome. This study, by appropriately adjusting for patient confounders and clustering of observations at the hospital-level is novel in its approach and clearly demonstrates the importance of hospital characteristics.

It should be noted, however, that this study did have some limitations. The small number of hospitals sampled prevented us from carrying out a multi-level model which would have enabled us to calculate direct estimates of hospital-level effects. Furthermore, the small event numbers per hospital led to wide confidence intervals and difficulty in interpreting our results for most hospitals. This limitation arose because the sample size calculation for the ASCNES study was arrived at partly pragmatically due to its exploratory nature, and was based on the primary outcome of one-year mortality. Therefore, it did not account for the low event size of this secondary outcome, and the exclusion criteria of this present study. A larger sample size would have allowed us to statistically test for the interaction between hospital size and hospital stroke volume in influencing nursing home placement. As service characteristics were self-reported by hospital managers and service providers, information bias may also have been introduced. Moreover, we were unable to adjust for stroke severity and other established risk factors of institutionalization after strokes, such as ethnicity [35,36] and socio-economic status. However, we adjusted for proxy variables of stroke severity, such as whether the patient had a TACS [14] or showed symptoms of brain lateralization. Additionally, the region where the study was conducted serves mainly a white British Caucasian population [37], and hence any potential confounding by race will have been limited. Furthermore, although we used an aggregate measure of socio-economic status and did not find an obvious pattern in our graphical exploration of hospital factors, we cannot be sure whether this resulted in some residual confounding.

Finally, advances in stroke management over the past decade have occurred since the collection of the ASCNES data, and this has likely led to differences in the discharge destinations of patients. However, variations in rates of nursing home placements still exist between hospitals [2], which may be partly due to differences in hospital characteristics. Our study supports that variations in patient outcomes are influenced, in part, by hospital characteristics. This is still relevant despite changes in acute therapy and post-acute care, and thus should be explored further in the current context.

## 5. Conclusions

Overall, this study furthers the knowledge of nursing home discharges after strokes, an important outcome for patients, relatives, and health care providers, by illustrating that hospital characteristics do seem to play an important role in driving variations between hospitals. Further confirmatory work could lead to the development and implementation of health services research in this area.

## Figures and Tables

**Figure 1 healthcare-08-00390-f001:**
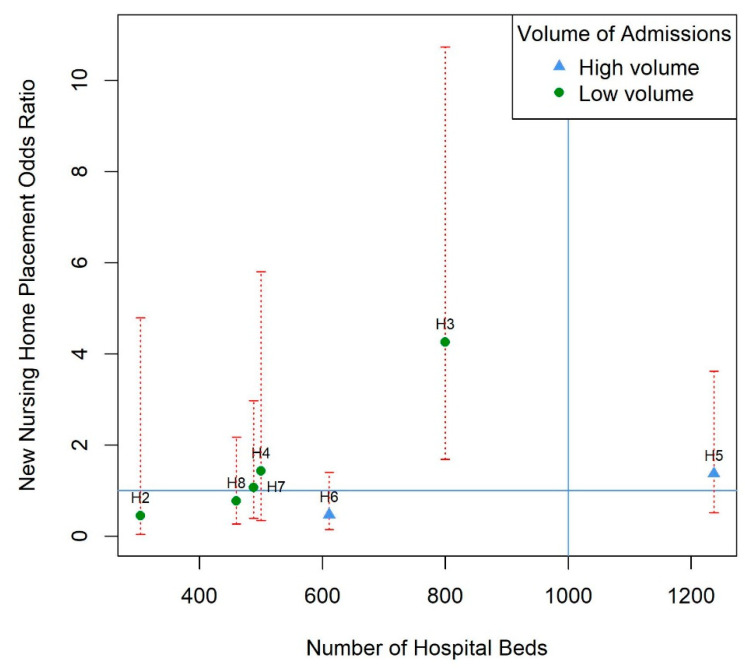
Model estimates of new nursing home placement odds ratio for each hospital against the size of the hospital (total number of hospital beds) and grouped by hospital stroke volume with 95% confidence intervals (red dotted lines). Volume is categorized into high (more than 50 admissions per month) and low (less than 50 admissions per month). Horizontal line represents an odds ratio of 1 for reference hospital 1. Vertical line represents the hospital size in hospital 1. This line is blue because the volume of stroke admissions is more than 50 admissions per month for this hospital. Multivariable regression model was adjusted for pre-stroke mRS score, discharge mRS score, total anterior circulation stroke (TACS), myocardial infarction or ischemic heart disease, complication, brain lateralization, and acute hospital length of stay, after multiple imputations for missing covariate data. H2–H8: hospital 2 to 8, consecutively.

**Table 1 healthcare-08-00390-t001:** Frequency of patients with stroke admitted and discharged to a nursing home placement (NHP) per hospital, and unadjusted odds ratio (OR) and 95% confidence intervals (CI) for new nursing home placement according to hospital (*n* = 1335).

Hospital	*n* (%) Stroke Patients Admitted during Study Period	Frequency of New NHP		Unadjusted Analysis		Adjusted Analysis ^1^	
		*n*	%	OR (95% CI)	*p*	OR (95% CI)	*p*
**1**	211 (16)	12	6	Reference	Reference	Reference	Reference
2	10 (1)	1	10	1.84 (0.22 to 15.76)	0.58	0.45 (0.04 to 4.79)	0.51
3	202 (15)	38	19	3.84 (1.94 to 7.59)	<0.001	4.26 (1.69 to 10.73)	0.002
4	63 (5)	8	13	2.41 (0.94 to 6.19)	0.07	1.43 (0.35 to 5.80)	0.61
5	328 (25)	22	7	1.19 (0.58 to 2.46)	0.63	1.37 (0.52 to 3.62)	0.52
6	168 (13)	15	9	1.63 (0.74 to 3.57)	0.23	0.47 (0.15 to 1.40)	0.17
7	183 (14)	20	11	2.03 (0.97 to 4.29)	0.06	1.07 (0.39 to 2.97)	0.89
8	170 (13)	19	11	2.09 (0.98 to 4.43)	0.06	0.77 (0.27 to 2.17)	0.62

^1^ Multiple logistic regression results adjusted for pre-stroke modified Rankin Scale (mRS) score, discharge mRS score, total anterior circulation stroke, history of myocardial infarction or ischemic heart disease, inpatient complication, and acute hospital length of stay.

**Table 2 healthcare-08-00390-t002:** Results of the sensitivity analyses with adjusted odds ratio (OR) and 95% confidence intervals (CI) for new nursing home placement according to hospital presented.

Hospital	Sensitivity Analysis ^1^		Sensitivity Analysis ^2^	
	OR (95% CI)	*p*	OR (95% CI)	*p*
**1**	Reference	Reference	Reference	Reference
2	0.24 (0.02 to 2.62)	0.24	-	-
3	4.26 (1.59 to 11.41)	0.004	4.33 (1.72 to 10.94)	0.002
4	2.88 (0.64 to 12.84)	0.17	1.43 (0.35 to 5.77)	0.62
5	1.29 (0.46 to 3.63)	0.63	1.38 (0.52 to 3.67)	0.51
6	0.56 (0.18 to 1.81)	0.34	0.48 (0.16 to 1.46)	0.20
7	0.97 (0.34 to 2.80)	0.96	1.11 (0.40 to 3.07)	0.84
8	0.70 (0.23 to 2.11)	0.53	0.80 (0.28 to 2.26)	0.67

^1^ Multiple logistic regression sensitivity analysis adjusted for age, sex, dementia, hemorrhagic stroke, pre-stroke modified Rankin Scale (mRS) score, discharge mRS score, total anterior circulation stroke, comorbidities (including dementia, previous stroke, diabetes mellitus, history of myocardial infarction or ischemic heart disease), prior antiplatelet or anticoagulant use, inpatient complication, and acute hospital length of stay, using multiple imputed dataset (*n* = 1335). ^2^ Multiple logistic regression sensitivity analysis excluding hospital 2, using multiple imputed dataset (*n* = 1325).

**Table 3 healthcare-08-00390-t003:** Hospital characteristics per individual hospital.

Characteristics	1	2	3	4	5	6	7	8
General Characteristics								
Catchment population	400,000	160,000	350,000	230,000	680,000	300,000	240,000	275,000
Hospital type	Tertiary	Secondary	Secondary	Secondary	Tertiary	Secondary	Secondary	Secondary
No. of ASCNES admissions per month	52	13	46	19	88	57	35	31
Facilities and Services								
No. of hospital beds	1000	304	800	500	1237	611	488	460
No. of stroke unit beds (per 100 admissions)	71	77	54	138	41	55	83	65
No. of hospital beds per CT scanners	500	304	400	250	518	306	244	230
Distance to vascular surgery (miles)	0	18	0	25	0	0	43	30
Distance to neurosurgery (miles)	0	18	58	89	61	38	48	30
Rehabilitation beds in the stroke unit	Yes	Yes	No	Yes	Yes	Yes	Yes	Yes
Early Supported Discharge provision	No	Yes	No	Yes	Yes	Yes	No	No
Stroke Unit Staffing Levels ^1^								
Senior doctors ^2^	0.34	0.25	0.49	0.47	0.42	0.31	0.62	0.87
Junior doctors ^2^	0.55	0.65	0.72	0.59	0.56	0.64	0.12	0.25
Health care associates and nurses (band 5–7)	9.2	8	6	7.4	7	5.3	6.5	10
Physiotherapists (band 2–8)	0.55	1	0.79	0.4	0.91	0.78	0.69	1
Occupational therapists (band 3–8)	0.49	0.5	1.4	0.59	0.6	0.58	0.52	1.1
Speech and language therapists	0.39	0.15	0.2	0.18	0.35	0.03	0.26	0.1
Dieticians								
No. of non-stroke patients treated daily on stroke unit ^1^	0.27	0	0.10	0.47	0.05	0.31	0.17	0
No. of stroke patients treated daily on other wards ^1^	0.14	1.25	0	0.30	0.01	0.41	0	0

^1^ Number of fte staff per five stroke unit beds (weighted average for the four study periods taken) ^2^ Weekday numbers only.

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
