# Peer review of "Variations in Rates of Discharges to Nursing Homes after Acute Hospitalization for Stroke and the Influence of Service Heterogeneity: An Anglia Stroke Clinical Network Evaluation Study"

_healthcare, 2020, doi:10.3390/healthcare8040390_

Round 1

Reviewer 1 Report

Summary:

This is prospective multicenter study on acute stroke patients which was performed to describe the factors influencing the hospital variability with regards to discharge placements to nursing homes. The authors report that 10% of discharges to nursing homes from their study overall and that increasing hospital sizes correlated with more nursing home placements, although this was also influenced by the number of stroke admissions.

Comments by section

Introduction

Could the authors comment on who makes the decision for a patient to be discharged to nursing home versus home or another destination, such as rehab? Is it the therapists, physicians, joint decision etc…? It would be interesting to know given the study is comparing factors, such as number of therapists available, number of physicians per five beds

Methods

  • Can the authors describe their rationale for using data from 2009-2011? Over the last decade significant changes to stroke management have occurred across the world both from an acute therapy and post stroke care standpoints and so these could play an important role in the outcomes of patients including discharge disposition. Considering the above, the data presented might be limited with regards to applicability in the current day and age.

  • Page 2 lines 69-71: Inclusion of only patients with a neurological impairment lasting > 24 hours can be considered a limitation which should be specified in the limitations section since, a proportion of patients might not have focal deficits past 24 hours and still have radiologically evidenced ischemic stroke especially patients who receive acute reperfusion therapies. Taking out these patients from the study could have falsely inflated the discharge disposition to nursing facilities or other rehab places.

  • Page 2 72-74

Interesting method of selecting patients (every 3rd month). While it was verified in a prior study, there do seem to be some limitations including decreased overall power due to smaller sample size.

  • Page 2 Lines 91-94, the authors state “We adjusted for patient risk factors of institutionalization (identified a priori from a literature review) which were unevenly distributed amongst the hospitals (P-value set at 0.05 significance level). These included a patient’s pre-stroke modified Rankin Scale (mRS) score, discharge mRS score, acute hospital length of stay, and whether the patient had a total anterior circulation stroke (TACS), history” This would be one of the most important limitations of the study since the adjustment was performed taking into considerations the reported literature review that the authors performed. Two comments pertain to this:
  1. Can they provide references for their literature review pertinent to these variable inclusions?
  2. Variables such as baseline NIHSS (stroke severity), age, gender, reperfusion therapy in ischemic stroke (detailed in point c), comorbidities and the type of stroke (hemorrhagic versus ischemic) have been consistently shown to significantly affect the discharge outcomes/institutionalization which were not considered in this study and can potentially confound the results of the study. Can the authors provide the results after adjustment for these important confounders?
  3. In terms of patient factors, there’s no mention of the kind of acute therapy patients were receiving. Were there some patients receiving intravenous thrombolytics or procedures, such as endovascular thrombectomy for ischemic stroke? As stated prior, these are patient presentation factors that could greatly affect outcome and discharge destination. Additionally, it may be important to consider/adjust for the percentage of patients who had ischemic versus hemorrhagic strokes as a patient factor and how that affected discharge destination as the type of stroke could play an important role in outcome. Both of these patient presentation factors could confound analysis of hospital characteristics that influence discharge destination. (Briefly addressed in Discussion 235)

  • Methods 95

The risk factors are all related to patient presentation and hospital stay except history of myocardial infarction/ischemic heart disease. Could the authors elaborate on how a prior myocardial infarction is considered a risk factor for institutionalization and/or provide a citation for this?

  • Methods 96

Could the authors potentially provide a table of examples of inpatient complications? It would be interesting to see the variation and degree of severity of complications.

Results

  • Results 152-153

How is having a neurosurgical facility on site (for thrombectomy?) and onsite vascular surgery a resource that may influence discharge destination? Could the authors elaborate why they included this in the hospital characteristics?

  • Results 169

Addressed, but wide confidence interval. These trends would be stronger if the study had better/adequate power.

  • Results 181

Brain lateralization is included as part of the criteria that were adjusted for when comparing new nursing home placement odds ratio to number of hospital beds but was not mentioned earlier as a factor that was adjusted for (lines). Could the authors elaborate why this one criteria was different between the two

Discussion

  1. Discussion 185-186, 191-193

These sentences could be combined as it is repetitive

  1. Discussion 209-210

Hospitals 1 and 5, while tertiary centers, had many more hospital beds to CT scanner ratio. If pressure on CT scanners does contribute to delay and subsequently affects discharge destination, do you have any thoughts on why this is not the case in tertiary centers?

  1. Discussion 230-233

The sample size could have been improved by using all patients in the study time frame rather than selecting 1 month of patients for every 3 months? Any clear rationale on this pattern of patient selection?

  1. Discussion 235-236

The authors state” Moreover, we were unable to adjust for stroke severity and other established risk factors of institutionalization after stroke, such as ethnicity” Can the authors state why they were unable to adjust these important risk factors? As delineated above in the methods, they play a very important role in the ultimate outcomes and discharge disposition and without such adjustment the strong inferences provided from the results are unfortunately limited.

Overall:

While the prospective nature of the study is a definite strength, the relatively smaller event size is a limitation. Additionally, multiple variables including and not limited to stroke severity, stroke type, and acute management etc. were not adjusted for and each of these are variables that could highly influence discharge destination and confound analysis of hospital-related factors and their relation to discharge destination.

Author Response

Summary:

This is prospective multicenter study on acute stroke patients which was performed to describe the factors influencing the hospital variability with regards to discharge placements to nursing homes. The authors report that 10% of discharges to nursing homes from their study overall and that increasing hospital sizes correlated with more nursing home placements, although this was also influenced by the number of stroke admissions.

Response: We thank the reviewer for their time in reviewing our paper and for their helpful comments. We provide responses to specific comments below. 

Comments by section

Point 1: Introduction

Could the authors comment on who makes the decision for a patient to be discharged to nursing home versus home or another destination, such as rehab? Is it the therapists, physicians, joint decision etc…? It would be interesting to know given the study is comparing factors, such as number of therapists available, number of physicians per five beds

Response 1: Thank you for this comment. Decisions on the patient’s discharge destination is arrived at collaboratively, following discussions with the patient, the patient’s family, and the stroke team (including the leading physician and allied healthcare professionals, such as social work). We agree that adding this information to the introduction would be useful for the reader. We have therefore inserted the following sentence on page 2 in the Introduction:

Lines 51-54: “In UK hospitals, the decision as to the patient’s discharge destination following stroke is arrived at collaboratively following input from the patient, the patient’s family, the leading physician, social work, and other allied health professionals.”

Methods

Point 2: Can the authors describe their rationale for using data from 2009-2011? Over the last decade significant changes to stroke management have occurred across the world both from an acute therapy and post stroke care standpoints and so these could play an important role in the outcomes of patients including discharge disposition. Considering the above, the data presented might be limited with regards to applicability in the current day and age.

Response 2: Thank you for the important point you make. The ASCNES research team designed the ASCNES study during this time period, and subsequently published their findings regarding their primary outcome of interest – one-year mortality (Myint et al. 2017). As the ASCNES dataset has a wealth of data on both patient and hospital characteristics, the current study made use of this to explore the impact of hospital on nursing home placement (NHP) rates. We believe this is a rich dataset that allowed us to answer our research questions. However, we do acknowledge that over the last decade there has been substantial changes to stroke management in terms of acute therapy and post-stroke care. For example, we recognise that early supported discharge has become more widespread in the UK, and that this, along with improvements in acute therapy could lead potentially to less people requiring nursing home care. However, because our research is concerned more about how hospital factors e.g. size, volume, and staffing influence this outcome, and as differences in hospital characteristics have remained, we believe our findings are still relevant. It is these differences in outcomes caused by hospital factors that we are highlighting, rather than the changes over time in patients being sent on to nursing home care caused by universal changes in stroke management.

We do take this comment on board, and have added the following to our Discussion limitation section:

Lines 287-292: “Finally, advances in stroke management over the past decade have occurred since the collection of the ASCNES data, and this has likely led to differences in the discharge destinations of patients. However, variations in rates of nursing home placements still exist between hospitals [2], which may be partly due to differences in hospital characteristics. Our study supports that variations in patient outcomes are influenced, in part, by hospital characteristics. This is still relevant despite changes in acute therapy and post-acute care, and thus should be explored further in the current context.”

Point 3: Page 2 lines 69-71: Inclusion of only patients with a neurological impairment lasting > 24 hours can be considered a limitation which should be specified in the limitations section since, a proportion of patients might not have focal deficits past 24 hours and still have radiologically evidenced ischemic stroke especially patients who receive acute reperfusion therapies. Taking out these patients from the study could have falsely inflated the discharge disposition to nursing facilities or other rehab places.

Response 3: Thank you for comment. We apologise for any confusion with our wording. We used the World Health Organization definition of stroke in order to exclude diagnoses of transient ischaemic attacks, subdural haematomas, and subarachnoid haemorrhage (Aho e al., 1980; Johnson et al., 2019). Stroke, defined as intracerebral ischaemic or haemorrhagic, were diagnosed by an accredited stroke physician and coded using International Classification of Diseases-10 (ICD-10). Patients who were thrombolysed were included the patient sample. We acknowledge that a small proportion of patients with ischaemic stroke who did not have focal deficits after 24 hours may have been missed and not included in the sample, but we believe this scenario may only have occurred in a small proportion of patients, and as we had a large overall sample of patients, we do not think this would have had a significant impact on our findings. We have edited our Methods section 2.2 Participants to provide better clarity to the reader:

Lines 73-74: Diagnoses by the stroke physician were coded using International Classification of Diseases-10 (ICD-10) and were confirmed through cerebral imaging.

Lines 76-78: This definition was used in order to exclude diagnoses of transient ischaemic stroke, subdural hematomas, and subarachnoid hemorrhages.  

References

Aho, K., Harmsen, P., Hatano, S., Marquardsen, J., Smirnov, V.E., Strasser, T. Cerebrovascular disease in the community: Results of a WHO collaborative study. Bull World Health Organ. 1980;58(1):113-130.

Johnson, C.O., Nguyen, M., Roth, G.A. et al. Global, regional, and national burden of stroke, 1990-2016: A systematic analysis for the global burden of disease study 2016. Lancet Neurol. 2019; 18(5):439-458.

Point 4: Page 2 72-74

Interesting method of selecting patients (every 3rdmonth). While it was verified in a prior study, there do seem to be some limitations including decreased overall power due to smaller sample size.

Response 4: Thank you for your comment. As mentioned, this selection method was validated in a previous methods study (Kwok et al., 2014). The sample size for the ASCNES study was arrived at partly pragmatically since it was exploratory in design, and it was determined for the primary outcome of interest – one-year mortality. The planned sample size was 2400 for this outcome, with the overall sample size for ASCNES being 2477. Therefore, the method of selecting patients every 3 months was not considered limiting on sample size for this outcome. We chose this method of selecting patients as it was a way to be more time, resource and cost efficient without compromising the integrity our results in relation to yearly case-mix and seasonal fluctuations.

However, we do recognise that because this current study excluded patients who died or who were previously resident in a nursing home, and the low event size of new nursing home placement that this meant the power to detect a significant effect was lowered. Perhaps, this could have been alleviated by increasing sample size either through selecting patients every month rather than every 3 months, or equally, by selecting patients every 3 months but over a longer time period. We therefore, do not necessarily think that the patient selection method caused this limitation, rather it was the sample size calculation at the outset of the study which was based on the primary outcome of interest that was the source of this limitation. We have added the following to our manuscript to underline and explain this limitation more explicitly, as you have rightly highlighted this important point. We have inserted the following clarifying sentence in section 2.1 Study design:

Lines 67-69 : “The ASCNES primary outcome of interest was one-year mortality, whereas discharge destination, the focus of this paper, was one of the secondary outcomes of interest.”

We also inserted to our Discussion limitation section the following sentence:

Lines 273-276: “This limitation arose because the sample size calculation for the ASCNES study was arrived at partly pragmatically due its exploratory nature, and was based on the primary outcome of one-year mortality. It therefore did not account for the low event size of this secondary outcome, and the exclusion criteria of this present study”.

Reference

Kwok CS, Musgrave SD, Price GM, Dalton G, Myint PK; Anglia Stroke Clinical Network Evaluation Study (ASCNES) Group. Similarity of patient characteristics and outcomes in consecutive data collection on stroke admissions over one month compared to longer periods. BMC Res Notes. 2014 Jun 6;7:342. Available at: https://bmcresnotes.biomedcentral.com/articles/10.1186/1756-0500-7-342

Point 5: Page 2 Lines 91-94, the authors state “We adjusted for patient risk factors of institutionalization (identified a priori from a literature review) which were unevenly distributed amongst the hospitals (P-value set at 0.05 significance level). These included a patient’s pre-stroke modified Rankin Scale (mRS) score, discharge mRS score, acute hospital length of stay, and whether the patient had a total anterior circulation stroke (TACS), history” This would be one of the most important limitations of the study since the adjustment was performed taking into considerations the reported literature review that the authors performed. Two comments pertain to this:

Point 5a: Can they provide references for their literature review pertinent to these variable inclusions?

Response 5a: We thank the reviewer for highlighting this. We have now provided a summary of our a priori literature search findings we conducted to identify patient risk factors of institutionalisation. These findings along with the pertinent references are presented in the Supplementary Materials (Supplementary Table S1). We have updated the text in the main script in section 2.4 Statistical Analysis to read (pg 3):

Lines 99-100: “Patient risk factors of institutionalization following stroke were identified a priori from a literature review (see Supplementary Table S1).”

Point 5b: Variables such as baseline NIHSS (stroke severity), age, gender, reperfusion therapy in ischemic stroke (detailed in point c), comorbidities and the type of stroke (hemorrhagic versus ischemic) have been consistently shown to significantly affect the discharge outcomes/institutionalization which were not considered in this study and can potentially confound the results of the study. Can the authors provide the results after adjustment for these important confounders?

Response 5b: We thank the reviewer for highlighting this and we apologise for the lack of clarity in the original manuscript. We agree that these variables are important patient risk factors for nursing home placement, and we identified these also from our a priori literature review (findings now included as Supplementary Table S1). Furthermore, most of these were collected in the ASCNES dataset. However, because the aim of the study was to examine whether variations in new NHP were partly explained by differences in hospital-level factors, we only considered (in our main model) the patient risk factors that were shown to be unevenly distributed (P≤0.05) between the hospitals. Those that were unevenly distributed were hence regarded as potential confounders between NHP (outcome of interest) and hospital ID/factors (independent variable of interest) and thereby included in the main model. We have amended section 2.4 Statistical Analyses to provide greater clarity on how we decided on patient risk factors adjustments (pg 3):

Lines 99-104: “Patient risk factors of institutionalization following stroke were identified a priori from a literature review (see Supplementary Table S1).The distribution of the patient risk factors available in the dataset was examined between hospitals using hypothesis testing. Factors which were unevenly distributed amongst the hospitals (P-value set at 0.05 significance level) were included in the main model as they were considered potential confounders of the relationship between our independent variable (hospital) and outcome of interest (institutionalization).”

We have also now provided an additional table in the supplementary file to show the distribution of all the patient risk factors we explored in relation to each hospital (Supplementary Table S2). This has been mentioned in the text:

Lines 104-108: “These included a patient’s pre-stroke modified Rankin Scale (mRS) score, discharge mRS score, acute hospital length of stay, and whether the patient had a total anterior circulation stroke (TACS), history of myocardial infarction or ischemic heart disease, or developed an inpatient complication (see Supplementary Table S2).”

As can be seen from this table, the distribution of patient age, gender, comorbidities, and the type of stroke were not statistically significantly different across the 8 hospitals, and hence not adjusted for in the main analysis. To note, we limited the number of variables added in our main model through this method as we felt this would help avoid the issue of overfitting.

Nonetheless, we did run a sensitivity analysis which included all these additional risk factors to check the robustness of our confounder selection procedure. This is mentioned under the Methods section 2.5 Sensitivity Analyses (pg 4), (now lines 151-154): We performed two sensitivity analyses to check the robustness of our findings. The first of these included all patient risk factors identified a priori in a literature review that were available in our dataset irrespective of whether they were evenly distributed between hospitals or not, and the second excluded hospital 2 which did not contribute data after January 2010.” We had previously presented the sensitivity analysis in Supplementary Tables S4 and S5 and reported on them with the brief sentence (pg4; lines 177-178): Findings were similar for the complete case and sensitivity analyses (Supplementary Tables S3, S4 and S5)”. We have now changed this by including the sensitivity analyses as a separate table in the main manuscript in order to provide clarity to the reader that these additional risk factors were considered, but did not impact on our interpretations and findings. This has been inserted as Table 2 at line 185. We have amended text to now read:

Lines 177-178: “Findings were similar for the complete case analysis (Supplementary Table S6) and sensitivity analyses (Table 2).”

With respect to baseline NIHSS (stroke severity), we did not include this in our model due to the level of missing data. A total of 73% cases had missing data on NIHSS, and we believe that this variable was only collected in patients that were eligible for thrombolysis. Therefore, we did not multiple impute this variable due to the potential for information bias. However, to reduce any residual confounding caused by the lack of adjustment for stroke severity, we included whether the patient had a total anterior circulation stroke (TACS), had brain lateralisation, and their mRS score at discharge as proxy measures. TACS and discharge mRS have both previously been shown to be correlated with stroke severity (Ghosh et al., 2018; Nedeltchev et al., 2005; Saver et al., 2012), and we hypothesised that patients with no symptoms of brain lateralisation are likely to be those with more severe stroke where the whole brain is affected. We have now explained this under 2.4 Statistical Analyses, pg 3:

Lines 110-117: “Furthermore, although we identified stroke severity as an important risk factor, 73% of cases had missing data on The National Institutes of Health Stroke Scale (NIHSS) score, and we believe that this variable was only collected in patients that were eligible for thrombolysis. Therefore, we did not multiply impute this variable due to the potential for introducing information bias. Instead, we adjusted for proxy variables of stroke severity, such as discharge mRS score [13], whether the patient had a TACS [14,15] or showed symptoms of brain lateralization. We hypothesized that patients with no symptoms of brain lateralization are likely to be those with more severe stroke where the whole brain is affected.”

We provide a specific response to the exclusion of reperfusion therapy below as this responds to point 5c also.

Point 5c: In terms of patient factors, there’s no mention of the kind of acute therapy patients were receiving. Were there some patients receiving intravenous thrombolytics or procedures, such as endovascular thrombectomy for ischemic stroke? As stated prior, these are patient presentation factors that could greatly affect outcome and discharge destination.

Response 5c: Thank you for highlighting this point and we apologise for not explaining this in our original manuscript submission. With respect to acute therapy treatments, including IV-tPA and endovascular thrombectomy, the reason we did not include these in our analysis was to avoid introducing overadjustment bias. We agree that these treatments will have an impact on discharge destination, however we regard that patient treatments such as these are “processes of care” measures which do not act as confounders but can instead be considered as intermediate (mediator) variables that lie on the causal pathway between the hospital-level covariates and stroke patient outcome. We have now added this explanation to the manuscript and refer to Schisterman et al., 2009 and Staplin et al., 2017 who provide an excellent account of why mediator variables shouldn’t be included when we are investigating the overall effect of a particular variable (in our case hospital factors, such as volume).

To provide an example of what is meant here, if we were to find that hospital type (tertiary vs secondary) was related to NHP rates we could hypothesize that one explanation for this finding is that in tertiary hospitals where access to specialised services and expertise is greater than secondary hospitals, the rate and quality of treatment provision with tPA is higher, which then affects the patient outcome. In other words, the fact that the hospital is tertiary could be the reason for increased provision/quality of treatment, which hence has an effect on outcome. If we were to include both variables in the analysis, the process measure (in this case, treatment provision) would dull/obscure any effect we see between hospital type and discharge destination, which could lead to an inaccurate conclusion that hospital type does not affect discharge destination. To support this further, we have now referenced a study by Tung et al., 2015 that shows that process of care measures such as use of different therapies and treatments, act as mediators of the relationship between hospital-level care (e.g. hospital type) and mortality after stroke. We have added the following in the methodology section (2.4 Statistical analyses; pg 3) to explain this more clearly in our manuscript:

Lines 118-121: “Processes of care measures such as the delivery of acute treatment therapies were not accounted for in our study as we believe they are mediator variables that lie on the causal pathway between hospital-level factors and stroke patient outcomes [16]. Their inclusion in our regression model could otherwise lead to over-adjustment bias by obscuring the overall effect of hospital [17,18].”

Point 5d: Additionally, it may be important to consider/adjust for the percentage of patients who had ischemic versus hemorrhagic strokes as a patient factor and how that affected discharge destination as the type of stroke could play an important role in outcome. Both of these patient presentation factors could confound analysis of hospital characteristics that influence discharge destination. (Briefly addressed in Discussion 235)

Response 5d: Thank you for highlighting this. As noted in response 5b, we do agree that stroke type is an important risk factor for discharge destination. However, the reason we did not include this variable in the main analysis is because we judged it would not be a confounding factor on the relationship between hospital and discharge destination as we have shown that this characteristic did not vary statistically across the hospitals. As mentioned above, we have now provided a supplementary table (Supplementary Table S2) to demonstrate this. We also ran a sensitivity analysis for all identified risk factors (including stroke type) irrespective of whether they were evenly distributed or not across the hospitals. Indeed, in our sensitivity analysis its association with NHP did not reach statistical significance, and it did not appear to affect the main findings of hospital variations. We have added the results of the sensitivity analysis to the main manuscript as Table 2 (on page 6). As noted above we also clarified this process in lines 99-104). To note, we haven’t presented the regression effect estimates per patient risk factor as our focus in the paper was on the effect of hospital.

 Point 6: Methods 95

The risk factors are all related to patient presentation and hospital stay except history of myocardial infarction/ischemic heart disease. Could the authors elaborate on how a prior myocardial infarction is considered a risk factor for institutionalization and/or provide a citation for this?

Response 6: We thank the reviewer for highlighting this point. We apologise for any oversight in not providing a reference for this in the manuscript. We considered prior myocardial infarction or presence of IHD as a comorbidity that may increase the risk of institutionalisation through the impact this comorbidity may have on the overall health of the patient. Having a stroke in addition to a prior MI or having IHD may increase chances of institutionalisation as these patients may already be receiving rehab for MI or are in generally ill health with their disease. Furthermore, this was considered as a potential patient level risk factor of NHP because previous studies have shown a significant association. For example, Tseng et al., 2015 showed in multivariable logistic regression that patients with heart disease had lower odds of being discharged to a nursing home compared to those who didn’t have heart disease. On the other hand, using the Dijon Stroke Registry, Bejot et al., 2012 demonstrated that history of heart failure was linked to an increased risk of admission to a nursing home after stroke. We therefore thought this was a comorbidity that ought to be investigated as a potential confounder. As noted in our response to point 5a above, we have now provided a summary table of our literature review findings in Supplementary Table S1.

Bejot, Yannick T, Odile G, et al. Poststroke disposition and associated factors in a population-based study: The Dijon Stroke Registry. Stroke. 2012;43(8):2071-2077.

Tseng HP, Lin FJ, Chen PT, et al. Derivation and validation of a discharge disposition predicting model after acute stroke. J Stroke Cerebrovasc Dis. 2015;24(6):1179-1186

Point 7: Methods 96

Could the authors potentially provide a table of examples of inpatient complications? It would be interesting to see the variation and degree of severity of complications.

Response 7: We thank the reviewer for their suggestion. We have now provided this in Supplementary Table S3 that outlines the different types of inpatient complications recorded, and their variation in frequency. The majority of inpatient complications were either recorded as pneumonia (22%), UTI (31%), or unspecified (i.e. patient was noted to have complication but not any of the pre-specified complications asked for; these included but were not limited to hospital-acquired infections and septicaemia) (33%). We regret to inform that data on the severity of complications was not collected.

Results

Point 8: Results 152-153

How is having a neurosurgical facility on site (for thrombectomy?) and onsite vascular surgery a resource that may influence discharge destination? Could the authors elaborate why they included this in the hospital characteristics?

Response 8: Thank you for your comment. Our hypothesis was that by having facilities such as neurosurgery and vascular surgery located onsite, patients in these hospitals (who require neurosurgical (e.g. hemicraniectomy) or vascular surgical interventions) would be treated earlier than those who would have to be triaged offsite. Patients who require such treatment but have to be transferred to a facility offsite may consequentially experience a delay in their treatment. Any delay in treatment of stroke can lead to poorer/lengthier recovery, and hence, we believed this may therefore mean a safe discharge home from hospital may be hindered for such patients. We therefore wanted to test this hypothesis by including these hospital factors in our exploratory analysis. To note, at the time of the study thrombectomy specifically was not in widespread use and is relatively rare in the UK setting today.

Point 9: Results 169

Addressed, but wide confidence interval. These trends would be stronger if the study had better/adequate power.

Response 9: We thank the reviewer for this comment, and would agree that the confidence intervals for hospital effects would be narrower if we had better power. In our response to point 4 above, we explained that the lack of power was largely due to the fact that the sample size for the ASCNES study was arrived at partly pragmatically since it was exploratory in design, and it was determined for the primary outcome of interest – one-year mortality. However, because this current study had a low event size and also had to exclude patients who died or who were previously resident in a nursing home, this meant the power to detect a significant effect was reduced. As outlined above we have added the following to our manuscript:

2.1 Study design (pg 2) lines 67-69: “The ASCNES primary outcome of interest was one-year mortality, whereas discharge destination, the focus of this paper, was one of the secondary outcomes of interest.”

We also inserted to our Discussion limitation section (pg 11) the following text:

Lines 273-276: “This limitation arose because the sample size calculation for the ASCNES study was arrived at partly pragmatically due its exploratory nature, and was based on the primary outcome of one-year mortality. It therefore did not account for the low event size of this secondary outcome, and the exclusion criteria of this present study”.

To note, despite this limitation, our regression model still showed a strong statistically significant difference in NHP placement in hospital 3 compared to hospital 1 (OR 4.26; 95% CI 1.69 to 10.73).

Point 10: Results 181

Brain lateralization is included as part of the criteria that were adjusted for when comparing new nursing home placement odds ratio to number of hospital beds but was not mentioned earlier as a factor that was adjusted for (lines). Could the authors elaborate why this one criteria was different between the two

Response 10: We thank the author for highlighting this, and apologise for our oversight. We have now added this to our methodology section, 2.4 Statistical analyses (pg 3):

Lines 114-115: Instead, we adjusted for proxy variables of stroke severity, such as discharge mRS score [13], whether the patient had a TACS [14,15] or showed symptoms of brain lateralization. We hypothesized that patients with no symptoms of brain lateralization are likely to be those with more severe stroke where the whole brain is affected.”

Discussion

Point 11: Discussion 185-186, 191-193

These sentences could be combined as it is repetitive

Response 11: We thank the reviewer for this point, and agree there is repetition here. In the first paragraph we wanted to provide a quick summary of the key findings of the study, and then follow this with a section on how each of our key findings relate to other works. This is the reason for the repetition. However, we would agree that changing sentence in lines 191-193 would minimise such a repetition in words. We have therefore changed this to read (pg 11):

Line 234: “Our main finding agrees with a previous study conducted across health regions in the UK [25]”.

We hope this has addressed this point.

Point 12: Discussion 209-210

Hospitals 1 and 5, while tertiary centers, had many more hospital beds to CT scanner ratio. If pressure on CT scanners does contribute to delay and subsequently affects discharge destination, do you have any thoughts on why this is not the case in tertiary centers?

Response 12: Thank you for highlighting this important point. It could be that both these hospital factors are important at influencing discharge destination (pressure on CT scanners causes a delay and increases risk of discharge destination; whilst tertiary centers, due to their more specialised expertise and facilities, lead to more effective treatment and hence better outcomes). The reason that hospital 1 and 5 who had very high numbers of hospital beds to CT scanners, did not have as high an odds of discharging patients to nursing homes, may therefore be because of the competing influencing factor of them being tertiary. In other words, the benefits of being a tertiary hospital may outweigh the impact that pressure on CT scanners have on discharge destination outcome. Therefore, I would suggest the observation you highlight is a reflection of the interplay of the different hospital characteristics influencing patient outcome.

Point 13: Discussion 230-233

The sample size could have been improved by using all patients in the study time frame rather than selecting 1 month of patients for every 3 months? Any clear rationale on this pattern of patient selection?

Response 13: Thank you for your comment. In our response to point 4 above, we explain that the lack of power was largely due to the fact that the sample size for the ASCNES study was arrived at partly pragmatically since it was exploratory in design, and it was determined for the primary outcome of interest – one-year mortality. The planned sample size was 2400 for this outcome, with the overall sample size for ASCNES being 2477. Therefore, the method of selecting patients every 3 months was not considered limiting on sample size for this outcome. We chose this method of selecting patients as it was a way to be more time, resource and cost efficient without compromising the integrity our results in relation to yearly case-mix and seasonal fluctuations. This was demonstrated in the study by Kwok et al., 2014. However, we do recognise that because this current study excluded patients who died or who were previously resident in a nursing home, and there was a low event size of new nursing home placement, this meant the power to detect a significant effect was lowered. Perhaps, this could have been alleviated by increasing sample size either through selecting patients every month rather than every 3 months, or equally, by selecting patients every 3 months but over a longer time period. We, therefore, do not necessarily think that the patient selection method caused this limitation, rather it was the sample size calculation at the outset of the study which was based on the primary outcome of interest. In short, as we based our data collection request on planned sample size for mortality, we overlooked the need for a larger sample for this secondary outcome. Requesting all patients in the study frame from the network would have entailed extra funding, resources, and time – unfortunately, barriers we could not overcome in hindsight. We have added the following to our manuscript to underline and explain this limitation more explicitly, as you have rightly highlighted this important point:

We have inserted the following clarifying sentence in methodology section 2.1 Study design (pg11):

Lines 67-69: “The ASCNES primary outcome of interest was one-year mortality, whereas discharge destination, the focus of this paper, was one of the secondary outcomes of interest.”

We also inserted to our Discussion limitation section (pg 11) the following sentences:

Lines 273-276: “This limitation arose because the sample size calculation for the ASCNES study was arrived at partly pragmatically due its exploratory nature, and was based on the primary outcome of one-year mortality. It therefore did not account for the low event size of this secondary outcome, and the exclusion criteria of this present study”.

References

Kwok CS, Musgrave SD, Price GM, Dalton G, Myint PK; Anglia Stroke Clinical Network Evaluation Study (ASCNES) Group. Similarity of patient characteristics and outcomes in consecutive data collection on stroke admissions over one month compared to longer periods. BMC Res Notes. 2014 Jun 6;7:342. Available at: https://bmcresnotes.biomedcentral.com/articles/10.1186/1756-0500-7-342

Point 14: Discussion 235-236

The authors state “Moreover, we were unable to adjust for stroke severity and other established risk factors of institutionalization after stroke, such as ethnicity” Can the authors state why they were unable to adjust these important risk factors? As delineated above in the methods, they play a very important role in the ultimate outcomes and discharge disposition and without such adjustment the strong inferences provided from the results are unfortunately limited.

Response 14: Thank you for your comment and highlighting the importance of stroke severity and ethnicity in determining NHP placement. Although we recognise that stroke severity has been shown to influence NHP placement and we did collect data on NIHSS score, as we explain in our response to point 5b above, we did not include this in our model due to the level of missing data and risk of information bias. We inserted the following paragraph to explain this in our methodology section 2.4 Statistical analyses (pg 3):

Lines 110-117: “Furthermore, although we identified stroke severity as an important risk factor, 73% of cases had missing data on The National Institutes of Health Stroke Scale (NIHSS) score, and we believe that this variable was only collected in patients that were eligible for thrombolysis. Therefore, we did not multiply impute this variable due to the potential for introducing information bias. Instead, we adjusted for proxy variables of stroke severity, such as discharge mRS score [13], whether the patient had a TACS [14,15] or showed symptoms of brain lateralization. We hypothesized that patients with no symptoms of brain lateralization are likely to be those with more severe stroke where the whole brain is affected.”

With regards to ethnicity and also acknowledging that studies have shown this to be a risk factor for NHP placement, we have not included this in our models because, unfortunately this variable was not collected. However, whilst we cannot provide exact ethnic mix at the patient-level, the region where the study was conducted serves mainly a white British Caucasian population, especially for the age group studied. Looking at the Office of National Statistics for 2011 for some insight into the ethnicity/racial distribution across this area it was seen that the three counties that are served by the 8 hospitals in our study do not differ greatly in their racial make-up. White race constitutes more than 90% in each of the counties, with Asian and Black race only making up a small proportion <3%, ≤2% respectively. To address this limitation, we noted in the manuscript at line 282-284 “Also, the region where the study was conducted serves mainly a white British Caucasian population [37], and hence any potential confounding by race will have been limited.”

Point 15: Overall:

While the prospective nature of the study is a definite strength, the relatively smaller event size is a limitation. Additionally, multiple variables including and not limited to stroke severity, stroke type, and acute management etc. were not adjusted for and each of these are variables that could highly influence discharge destination and confound analysis of hospital-related factors and their relation to discharge destination.

Response 15: Thank you for providing this summary of your key points and areas on which the study can improve. We feel through our re-submission of the manuscript, and the above responses we have provided to each specific point, we have attempted to address these sufficiently. We again thank you for your time.

Reviewer 2 Report

Thank you for interesting and higly valuable study for stroke clinicians and managers of stroke services.

Your findings that larger hospitals (size) are associated with poorer outcomes, in constrast with association of higher hospital stroke volume with better outcomes, are of particular interest. Would you plan a larger sample size study for more reliable statistical confirmation of these interactions between hospital size, hospital stroke volume and nursing home placement?

Your findings of the superiority of early stroke rehabilitation started in a comprehensive stroke unit is also very important, despite the lack of conclusive evidence of benefits of early mobilization after stroke in recent systematic review (Langhorne P et al, 2018).

I do not have any other major comments.

I have  

Author Response

Point 1: Thank you for interesting and highly valuable study for stroke clinicians and managers of stroke services.

Response 1: We thank the reviewer for their time, and their positive and helpful comments. We provide responses to specific comments below.

Point 2: Your findings that larger hospitals (size) are associated with poorer outcomes, in contrast with association of higher hospital stroke volume with better outcomes, are of particular interest. Would you plan a larger sample size study for more reliable statistical confirmation of these interactions between hospital size, hospital stroke volume and nursing home placement?

Response 2: Thank you for your comment and your query regarding a study with a larger sample size. Our plan was to use the Sentinel Stroke National Audit Programme (SSNAP) data (which, in its entirety, covers 192 acute care hospitals across England, Wales and Northern Ireland) to look into these interactions further and confirm our findings of this smaller subset of hospitals in the UK. This would allow us to overcome some of the limitations mentioned pertaining to small hospital sample size and help validate our findings.

Point 3: Your findings of the superiority of early stroke rehabilitation started in a comprehensive stroke unit is also very important, despite the lack of conclusive evidence of benefits of early mobilization after stroke in recent systematic review (Langhorne P et al, 2018).

I do not have any other major comment

I have  

Response 3: Thank you for your comment and highlighting the importance of this finding.

Reviewer 3 Report

The manuscript by Tornes et al provides a carefully considered small study into the possible reasons for a large disparity in care with potentially very important insights for patients, hospital administrators and healthcare providers. For the most part they clearly present their data without overstating and make appropriate conclusions which should be used to inform standard of care. One variable that could be further considered is the average socioeconomic background of each hospital catchment area - are they comparable? Is Hospital 3 an area where the elderly have less access to care in general? These are, however, small suggestions. Overall the manuscript is thoughtful and well written. 

Author Response

Point 1: The manuscript by Tornes et al provides a carefully considered small study into the possible reasons for a large disparity in care with potentially very important insights for patients, hospital administrators and healthcare providers.

Response 1: We thank the reviewer for their time, and their positive and helpful comments. We provide responses to specific comments raised by the reviewer below.

Point 2: For the most part they clearly present their data without overstating and make appropriate conclusions which should be used to inform standard of care. One variable that could be further considered is the average socioeconomic background of each hospital catchment area - are they comparable?

Response 2: Thank you for your comment and highlighting this. Unfortunately, the socioeconomic status of patients was not collected in our dataset. However, looking at the Office for National Statistics for the year 2010 the mean English Index of Multiple Deprivation (IMD) score for the three separate counties in which the 8 hospitals serve are relatively close (Suffolk: 16.23, Norfolk: 15.75, Cambridgeshire: 13.89). The higher the IMD score, the less the area is considered deprived. It could also be assumed therefore that admissions to the different hospitals do not vary substantially in relation to their SES. As suggested, we have now included this as one of the hospital-level factors to explore graphically. We have therefore added the following paragraph to the methodology section 2.4 Statistical Analyses:

Lines 134-142: “The index of multiple deprivation (IMD) score was also explored graphically to explore whether the socio-economic status of the hospital catchment area may explain any differences in outcome observed between hospitals. This is an aggregate measure and is based on several domains, including income, employment, education, health, crime, barriers to housing and services and the living environment, that are believed to provide an indication of deprivation. England is sub-divided into 32, 844 smaller areas, with an IMD score of 1 representing the area in England that is considered to be the most deprived and a score of 32, 844 the least deprived [21]. Here we assigned the mean 2010 IMD scores of the areas that make up the counties of Suffolk, Norfolk and Cambridgeshire and assigned these to each of the hospitals to which they are located [22].”

We have also inserted this as a Supplementary Figure (Figure S20). As can be seen, no discernable patterns were uncovered. We have acknowledged this in the Results section:

Line 209-210: “No obvious patterns emerged for any of the hospital characteristics in our graphical analysis (Supplementary Figures S3 to S20).

However, as IMD is a neighborhood aggregate measure it would be difficult to say conclusively that socioeconomic status does not confound our findings unless we explicitly include this in the model at the patient-level. We have therefore moderated our findings further by stating in our Discussion section:

Lines 279-281: “Moreover, we were unable to adjust for stroke severity and other established risk factors of institutionalization after stroke, such as ethnicity [35,36] and socio-economic status.”

And

Lines 284-286: “Furthermore, although we used an aggregate measure of socio-economic status and did not find an obvious pattern in our graphical exploration of hospital factors, we cannot be sure whether this resulted in some residual confounding.”

Point 3: Is Hospital 3 an area where the elderly have less access to care in general? These are, however, small suggestions. Overall the manuscript is thoughtful and well written. 

Response 3: Thank you for your positive comments and for enquiring about the accessibility of care in the different areas. Unfortunately, it was beyond the scope of this study to look at access to social care in the different areas. Indeed, the differences in nursing home placements (NHP) between the hospitals may in part reflect the differences in local access/availability of nursing home care. However, we have hypothesized that if access/availability of social services were the cause for hospital 3’s apparent higher NHP rate i.e. that this hospital is in an area where there are much more spaces and funding for nursing home care compared to hospital 1, this should probably be reflected when we compare all the different destinations stroke patients in the full dataset were discharged to. In other words, we would expect to see more patients in hospital 1 being sent to “interim” care than hospital 3, because they are waiting for a nursing home space rather than not needing a place. However, when looking at the distribution of discharge destination for all patients in the dataset, this was not the case (i.e. hospital 3 had more patients sent to interim care than hospital 1). This figure was mentioned in the Results section in lines 169-172: “To note, when examining the original study population (including those who were discharged to settings other than home or nursing homes) the hospitals with the lowest rates of nursing home placements were not the hospitals with the highest proportion of patients discharged to an interim or rehabilitation setting, or who died as an inpatient (Supplementary Figure S2).”, and is presented in the Supplementary File as Figure S2. However, we agree with the reviewer that this is an important point, and so believe that future research should explore this further.

Round 2

Reviewer 1 Report

Thank you for answering all the queries in detail.